# Survival and new-onset morbidity after critical care admission for acute pancreatitis in Scotland: a national electronic healthcare record linkage cohort study

Chiara Ventre,[1] Sian Nowell,[2] Catriona Graham,[3] Doug Kidd,[2] Christos Skouras,[1] Damian J Mole[1,4]

[1]Surgery, School of Medicine, University of Edinburgh, Edinburgh, UK
[2]Electronic Data Research and Innovation Service (eDRIS), NHS National Services Scotland, Edinburgh, UK
[3]Edinburgh Clinical Trials Unit, University of Edinburgh, Edinburgh, UK
[4]MRC Centre for Inflammation Research, University of Edinburgh, Edinburgh, UK

**Correspondence to**
Professor Damian J Mole;
damian.mole@ed.ac.uk

## ABSTRACT

**Introduction** Severe acute pancreatitis (AP) requiring critical care admission (ccAP) impacts negatively on long-term survival.

**Objective** To document organ-specific new morbidity and identify risk factors associated with premature mortality after an episode of ccAP.

**Design** Cohort study.

**Setting** Electronic healthcare registries in Scotland.

**Participants** The ccAP cohort included 1471 patients admitted to critical care with AP between 1 January 2008 and 31 December 2010 followed up until 31 December 2014. The population cohort included 3450 individuals from the general population of Scotland frequency-matched for age, sex and social deprivation.

**Methods** Record linkage of routinely collected electronic health data with population matching.

**Primary and secondary outcome measures** Patient demographics, comorbidity (Charlson Comorbidity Index), acute physiology, organ support and other critical care data were linked to records of mortality (death certificate data) and new-onset morbidity. Kaplan-Meier and Cox regression analyses were used to identify risk factors associated with mortality.

**Results** 310 patients with AP died during the index admission. Outcomes were not ascertained for five patients, and the deprivation quintile was not known for six patients. 340 of 1150 patients in the resulting postdischarge ccAP cohort died during the follow-up period. Greater comorbidity measured by the Charlson score, prior to ccAP, negatively influenced survival in the hospital and after discharge. The odds of developing new-onset diabetes mellitus after ccAP compared with the general population were 10.70 (95% CI 5.74 to 19.94). A new diagnosis of myocardial infarction, stroke, heart failure, liver disease, peptic ulcer, renal failure, cancer, peripheral vascular disease and lung disease was more frequent in the ccAP cohort than in the general population.

**Conclusions** The persistent deleterious impact of severe AP on long-term outcome and survival is multifactorial in origin, influenced by pre-existing patient characteristics and acute episode features. Further mechanistic and epidemiological investigation is warranted.

## Strengths and limitations of this study

► This study is on a large contemporary cohort of patients with acute pancreatitis (AP) covering a national population (Scotland).

► Through secure record linkage, postdischarge critical care AP morbidity data are analysed in the context of episode-specific and pre-existing morbidity data.

► The use of pre-existing national databases resulted in low, but not negligible, amounts of missing data.

► The amount of missing data might have been further reduced had it been possible to prospectively capture all primary data.

► Only gallstone aetiology could be specifically examined due to data inaccuracies in the recording of other aetiologies of AP, specifically alcohol excess.

► The analysis of existing and new comorbidities was limited by the relatively small proportion of patients affected by each comorbidity, and because comorbidities derived from Scottish Morbidity Record 01 data only reveal diagnoses made at the time of a hospital admission and therefore are an underestimate of the true population prevalence.

## INTRODUCTION

Acute pancreatitis (AP) is the most common gastrointestinal cause of emergency hospital admission. The incidence of AP is increasing, and in Scotland is 31.8 per 100 000.[1–5] The overall case fatality in AP is 5%.[5] Although most cases are mild and self-limiting, one in four patients with AP develops multiple organ dysfunction syndrome (AP-MODS) and requires critical care admission.[6] AP-MODS is the single most important determinant of death from AP,[7] with mortality in patients with AP-MODS reaching 21.7%.[5] Recently, we reported that AP-MODS has detrimental consequences even for those who survive the acute episode, who have a reduced overall

survival compared with AP without MODS.[8] Prevention of AP-MODS in humans remains an elusive goal,[9] and it is therefore important to characterise the lasting impact on survivors to help maximise their long-term well-being.

AP has many potential causes, of which gallstones and alcohol are most frequently implicated.[6 10] The resulting inflammatory reaction within the pancreas may become overamplified and precipitate a systemic inflammatory response, shock and organ dysfunction.[6 10–13] There is marked interindividual heterogeneity in the number of organ systems involved, and AP-MODS can affect any organ system, with the respiratory and renal systems most frequently affected.[14–17] Moreover, the severity of organ dysfunction is highly variable, and interventions including invasive ventilation and renal replacement therapy can be required for durations raging from 1 day to 10 weeks.[18 19] AP-MODS determines mortality during the index admission,[20] but it is not certain which organ-specific failures are particularly associated with deterioration to death. One study linked hepatic and renal failures with the highest mortality risk,[19] whereas another placed greater negative influence after failure of the cardiovascular, pulmonary and gastrointestinal systems.[16]

Importantly, it is not completely understood which specific organ deficits may persist in survivors of AP-MODS. AP-MODS has been associated with an increased incidence of diabetes in AP survivors,[15 21 22] and age and working status are important in predicting recovery of quality of life and functional capacity.[21] Moreover, given the heterogeneity of the course of AP-MODS, it is unclear if a subgroup of AP-MODS survivors is at particularly high risk of a poor outcome. In the absence of an intervention to prevent AP-MODS, a deeper understanding of the persistent pathophysiological impact left by AP-MODS is needed. Therefore, our aim in this study was to integrate routinely collected data to investigate the causes and predictors of mortality in the years following an episode of AP requiring critical care admission.

## METHODS

### Study design, data security and patient confidentiality

This retrospective cohort study was conducted in collaboration with Electronic Data Research and Innovation Service to facilitate record linkage from multiple national databases. Information governance and security protocols were adhered throughout the investigation. All primary data were stored securely. Individual informed consent was not required or sought for this study.

### Patient and public involvement

We work closely with our patient and public involvement group, APPLe (Acute Pancreatitis Patient Liaison), to develop our research projects and strategies. This study received general input from members of APPLe as part of a consortium building workshop for the APPreSci Consortium (Acute Pancreatitis Precision Science, www.appresci.com), but APPLe was not involved in the data collection, analysis or manuscript preparation.

### Patient identification and data collection

All data were handled according to the Charter for Safe Havens in Scotland.[23] The Scottish Intensive Care Society Audit Group (SICSAG) WardWatcher database[24] was used to identify all patients admitted to critical care with AP between 1 January 2008 and 31 December 2010. AP was defined as any admission to critical care where the primary diagnosis coded by the intensive care senior clinician on duty was recorded as International Classification of Diseases, 10th Revision (ICD-10) classification K85 (acute pancreatitis). Where an individual was admitted with AP on more than one occasion, the earliest AP episode was taken as the index episode. There were no additional exclusion criteria. We performed a record linkage analysis of the Scottish Morbidity Record (SMR) 01 (general acute inpatient and day cases), General Register Office death records, SICSAG (critical care) and Community Health Index (CHI) databases. Causes of death of the general population of Scotland were obtained from National Records of Scotland Vital Events Tables.[25] Patient outcomes were recorded from the date of their index admission until the end of the follow-up period on 31 December 2014. Those lost to follow-up were censored at the point of last known contact. Prior to analysis, data records were linked using unique patient identifiers in order to maintain confidentiality.

### Variables of interest

The primary outcome of interest was death. The secondary outcomes were cause of death and new-onset morbidity. The following details of the index AP episode were recorded for each patient: gallstone aetiology (from SMR01 data), APACHE II score (Acute Physiology and Chronic Health Evaluation score, V.2), length of stay in critical care, level of critical care admission (high-dependency unit [HDU] or intensive care unit [ICU]),[26] and the requirement for renal replacement therapy, invasive ventilation, non-invasive ventilation, continuous positive airways pressure or vasopressor support (all from SICSAG data). In addition, the following patient characteristics were recorded: age on admission (from SICSAG), gender (from the CHI database), Scottish Index of Multiple Deprivation, Charlson score for comorbidity (calculated from the SMR01 records for each patient in the 5 years prior to admission)[27] and the number of comorbid conditions contributing to the Charlson score. The cause of death was obtained for each deceased patient and sorted according to the ICD-10 code into one of five categories: cardiovascular/circulatory, respiratory, neoplasia, digestive/metabolic or other causes, as shown in online supplementary table 1.

### Statistical analysis

IBM SPSS Statistics V.21 was used for all analyses. Categorical variables were reported as the absolute frequency

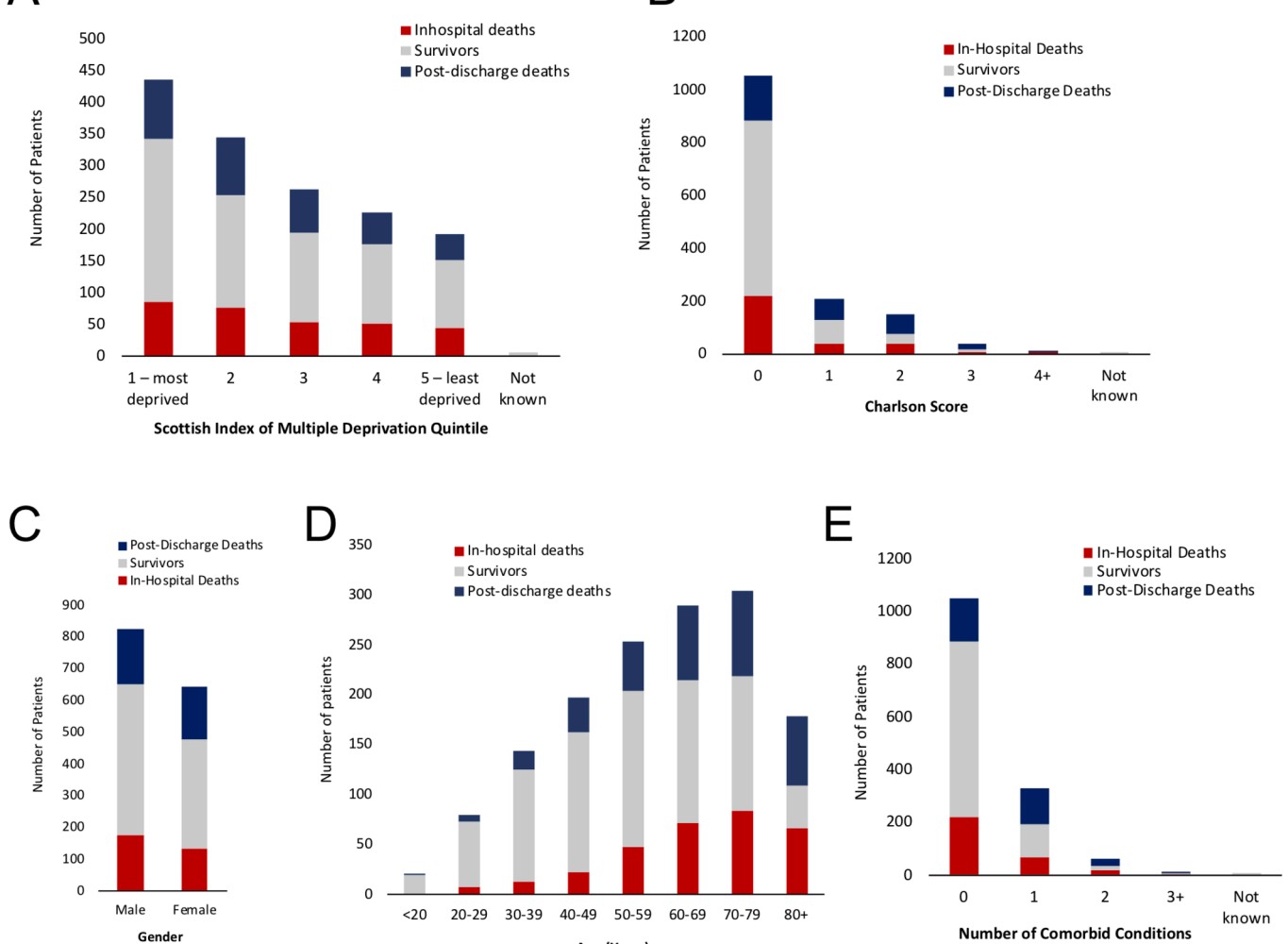

**Figure 1** Study population demographics. Visual representation of the demographic characteristics for the study cohort (n=1471 patients) with stacked bar charts. In all panels, the absolute number of patients per variable category is charted: in red are patients who died in the hospital, those who died postdischarge are in blue and those surviving to the end of follow-up are in grey. The following attributes of the cohort are depicted sequentially in each respective panel: (A) Scottish Index of Multiple Deprivation, (B) Charlson score, (C) gender, (D) age (transformed to a categorical variable) and (E) number of comorbid conditions contributing to the Charlson score.

and percentage. Continuous variables were reported as the mean±SD or the median±IQR. Kaplan-Meier analysis was used to demonstrate survival with respect to demographic and clinical factors, with the significance of differences assessed using the log-rank (Mantel-Cox) test. Survival was calculated as the time from the index admission to the hospital with AP to death; analyses and plots were done for the whole cohort, and for the cohort excluding those members who died during the index episode of AP in order to allow for analysis of long-term outcomes in survivors of the index episode, as specified in the Results section and in the figures.

The proportional hazards assumption was tested using log(-log) plots of the survival function over time, to confirm that the curves were approximately parallel. A multivariate Cox regression model was constructed to account for potential interactions between predictor variables. Covariates were added to the model using a forward stepwise method. At each step, the covariate found to be most significant was retained in the model. The threshold for retention in the models created using SPSS was p=0.01. After each addition, the covariates already present in the model were tested for removal depending on the significance of the likelihood ratio with and without each covariate.

A p value of 0.05 or less was considered significant. Where multiple pairwise comparisons were made—age group (<20, 20–29, 30–39, 40–49, 50–59, 60–69, 70–79, >80 years), Charlson score (0, 1, 2, 3, 4+), number of comorbid conditions (0, 1, 2, 3+)—the Bonferroni correction was applied to account for the quantity of comparisons being made.

### Secondary analysis of associated comorbidities

A control group was created from the general population using the CHI database register of all patients in

**Table 1** Demographic characteristics of the ccAP cohort

| | Died during index admission | % of total | Survived index admission | % of total | Died after hospital discharge | % of total |
|---|---|---|---|---|---|---|
| Gender | | | | | | |
| Male | 175 | 21 | 475 | 58 | 175 | 21 |
| Female | 135 | 21 | 341 | 53 | 165 | 26 |
| Age group | | | | | | |
| <20 | 0 | 0 | 19 | 95 | 1 | 5 |
| 20–29 | 7 | 9 | 66 | 84 | 6 | 8 |
| 30–39 | 13 | 9 | 112 | 78 | 19 | 13 |
| 40–49 | 22 | 11 | 141 | 72 | 34 | 17 |
| 50–59 | 47 | 19 | 157 | 62 | 49 | 19 |
| 60–69 | 72 | 25 | 143 | 49 | 75 | 26 |
| 70–79 | 83 | 27 | 135 | 44 | 86 | 28 |
| 80+ | 66 | 37 | 43 | 24 | 70 | 39 |
| Scottish Index of Multiple Deprivation quintile | | | | | | |
| 1—most deprived | 86 | 20 | 257 | 59 | 92 | 21 |
| 2 | 75 | 22 | 179 | 52 | 90 | 26 |
| 3 | 54 | 21 | 141 | 54 | 67 | 26 |
| 4 | 51 | 23 | 126 | 56 | 49 | 22 |
| 5—least deprived | 44 | 23 | 107 | 55 | 42 | 22 |
| Not known | 0 | 0 | 6 | 100 | 0 | 0 |
| Charlson Comorbidity Index | | | | | | |
| 0 | 219 | 21 | 666 | 63 | 167 | 16 |
| 1 | 41 | 20 | 91 | 44 | 77 | 37 |
| 2 | 38 | 25 | 41 | 27 | 71 | 47 |
| 3 | 7 | 18 | 12 | 32 | 19 | 50 |
| 4+ | 5 | 36 | 3 | 21 | 6 | 43 |
| Not known | 0 | 0 | 3 | 100 | 0 | 0 |
| Number of comorbid conditions contributing to the Charlson Comorbidity Index | | | | | | |
| 0 | 219 | 21 | 666 | 63 | 167 | 16 |
| 1 | 70 | 21 | 126 | 38 | 136 | 41 |
| 2 | 18 | 28 | 16 | 25 | 31 | 48 |
| 3+ | 3 | 21 | 5 | 36 | 6 | 43 |
| Not known | 0 | 0 | 3 | 100 | 0 | 0 |

The absolute number of patients and row percentages per each category for the following variables of interest are presented: gender, age group, Scottish Index of Multiple Deprivation,[43] Charlson Comorbidity Index[27] and the number of comorbid conditions contributing to the Charlson Comorbidity Index.
ccAP, critical care requiring acute pancreatitis.

Scotland. Controls were frequency-matched on deprivation quintile, age (by year of birth) and sex. Three controls were selected for every member of the exposed group. Comorbidities at index admission for AP were obtained from the SMR01 computerised acute hospital discharge records (day cases or inpatients) in the 5-year look-back period from date of admission for the index AP episode. Comorbidities that developed after discharge were ascertained from admissions after the index admission for AP up to 31 December 2014. The comorbidities that developed after the index event were then compared in the exposed and unexposed groups with a two-sample z test. We calculated the OR (and 95% CI) of developing each comorbidity given previous admission for AP needing critical care, compared with people with no previous AP admission.

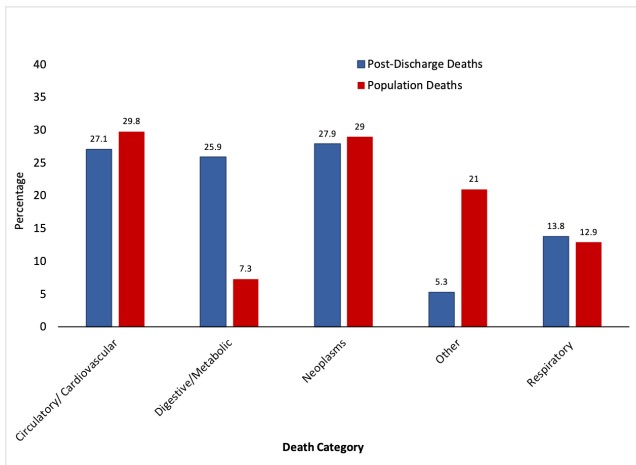

**Figure 2** Comparison of the causes of death between the ccAP cohort and the general population. Bar chart of the causes of death of the postdischarge ccAP cohort (341 deaths—blue colour) and of those of matched controls from the general population of Scotland (381 060 deaths—red colour). The causes of death have been grouped into one of the following categories, according to the ICD-10 code: cardiovascular/circulatory, respiratory, neoplasia, digestive/metabolic or other causes (online supplementary table 1). ccAP, acute pancreatitis requiring critical care admission; ICD-10, International Classification of Diseases, 10th Revision.

## RESULTS
### Follow-up and survival
Between 1 January 2008 and 31 December 2010, 1471 patients were admitted to HDU or ICU with AP. The length of the follow-up period ranged from 4.0 to 7.0 years. The median duration of follow-up from the date of index admission for all patients with AP was 4.4 years (IQR 0.6–5.6 years) and 4.9 years (IQR 4.0–5.8 years) when patients who died in the hospital during the index admission were excluded. Sixteen patients moved to another country and were censored at the point of last known contact. Figure 1 outlines the demographics of the study population. Demographic data for the cohort are presented in table 1.

During the follow-up period, 651 of 1471 (44.3%) patients died. Of 651 deaths, 310 (47.8%) occurred during the index admission; the outcome of 5 was not known and the deprivation quintile for 6 other patients was not known. The postdischarge critical care AP (ccAP) cohort therefore included 1150 patients, of whom 340 died during the follow-up period. As over half of the study cohort survived to the end of follow-up, the median survival time could not be determined. The mean (±SD) survival time in the whole cohort was 4.4±0.1 years and 5.6±0.1 years once in-hospital deaths were excluded.

### Cause of death
In the postdischarge ccAP cohort, neoplasms were the leading cause of death (27.9%), followed by cardiovascular (27.1%) and digestive/metabolic deaths (25.9%) (figure 2). Other causes contributed only 5.3% of the total. This contrasted with the general population of Scotland for which a lower proportion of deaths were attributed to digestive causes (7.3%), while a markedly greater proportion of the general population controls were due to other causes (21.0%) (figure 2).

### Predictors of mortality in the postdischarge cohort
Independent negative risk factors for long-term survival included age (online supplementary figure 1), a Charlson score of 1 or greater (table 2 and figure 3A), and the number of comorbid conditions contributing to the Charlson score (table 2 and figure 3B). Survival did not differ significantly with the degree of social deprivation (online supplementary figure 2). Female gender was associated with a shorter survival on univariate analysis, but gender as a risk factor on the multivariate analysis was not significant (table 3 and figure 3C). Gallstone aetiology was associated with a lower mortality after discharge (table 3). Comparison with analyses that included in-hospital deaths indicated that these differences emerged postdischarge (online supplementary figures 1–6). Multivariate Cox regression analysis also identified increased age group and Charlson comorbidity score as poor prognostic factors (table 3).

A significant relationship was observed between the requirement for renal replacement therapy, respiratory or circulatory support and an increased risk of death when in-hospital deaths were included (online supplementary figures 7–11). However, no correlation between mortality and any of the aforementioned medical interventions, or gallstone aetiology, was observed when considering only postdischarge outcomes (online supplementary figures 6–11). Long-term survival of those who survived the index episode was significantly better where the length of stay in critical care during the index episode exceeded 20 days, compared with admissions of 0–4 or 10–19 days (figure 4A). The critical care setting was important—patients with AP-MODS admitted to ICU, and who survived that event, had better long-term survival compared with those who were admitted to HDU and survived (figure 4B,C and table 2).

### Development of new specific comorbidities
Patients in the ccAP cohort were significantly more likely to develop a range of cardiovascular, gastrointestinal, pulmonary and neoplastic conditions than matched controls (table 4). A particularly high risk of developing new-onset diabetes was noted (OR 10.70, 95% CI 5.74 to 19.94), with 3.9% of the ccAP cohort developing new diabetes during the follow-up period compared with 0.4% of the matched control group. The risk of developing renal disease requiring hospital admission was also markedly increased (OR 9.15, 95% CI 2.95 to 28.43), but whether this was confounded by new or existing diabetes could not be ascertained, and the number of people affected by renal disease was small in both cohorts. The risks for developing other comorbidities are presented in table 4.

In addition to evaluating the risk of new-onset comorbidity, we examined whether the baseline comorbidities

**Table 2** Predictors of long-term mortality—univariate regression analysis

| Risk factor | | n | HR | 95% CI | P value |
|---|---|---|---|---|---|
| Age | Under 20 | 20 | – | | |
| (reference <20 years) | 20–29 | 72 | 1.6 | 0.2 to 13.1 | 0.674 |
| | 30–39 | 131 | 2.9 | 0.4 to 21.8 | 0.296 |
| | 40–49 | 175 | 3.9 | 0.5 to 28.5 | 0.180 |
| | 50–59 | 206 | 5.0 | 0.7 to 36.5 | 0.110 |
| | 60–69 | 218 | 7.9 | 1.1 to 57.0 | 0.040 |
| | 70–79 | 221 | 8.7 | 1.2 to 62.4 | 0.032 |
| | 80+ | 113 | 17.3 | 2.4 to 124.4 | 0.005 |
| Gender | Male | 650 | – | | |
| (reference male) | Female | 506 | 1.2 | 1.0 to 1.5 | 0.049 |
| Charlson score | 0 | 833 | – | | |
| (reference 0) | 1 | 168 | 2.6 | 2.0 to 3.4 | <0.001 |
| | 2 | 112 | 4.7 | 3.6 to 6.2 | <0.001 |
| | 3 | 31 | 3.8 | 2.4 to 6.1 | <0.001 |
| | 4 or more | 9 | 5.3 | 2.4 to 12.0 | <0.001 |
| Number of comorbid conditions | 0 | 833 | – | | |
| (reference 0) | 1 | 262 | 3.2 | 2.6 to 4.0 | <0.001 |
| | 2 | 47 | 4.5 | 3.0 to 6.7 | <0.001 |
| | 3 | 11 | 3.3 | 1.5 to 7.5 | 0.004 |
| Length of stay | Less than 20 days | 1079 | – | | |
| (reference <20 days) | Longer than 20 days | 77 | 0.4 | 0.2 to 0.8 | 0.006 |
| Level of critical care | ICU | 251 | – | | |
| (reference ICU) | HDU | 905 | 1.2 | 1.0 to 1.4 | 0.019 |

The HR, 95% CI and p value of Wald's test are presented for each variable found to significantly affect postdischarge survival on univariate regression analysis. P<0.05 was considered significant. The reference category for each variable is appended. Age has been transformed to a categorical variable for the purposes of the analysis.

ccAP, acute pancreatitis requiring critical care admission; HDU, high-dependency unit; ICU, intensive care unit; n, number of patients per category.

of the population who experience an episode of AP might be different from the general population (table 4). At the time of presentation with their index episode, patients with AP needing critical care were significantly more likely to have existing comorbidities that included cardiac, lung, renal or peripheral vascular disease, heart failure, connective tissue disorders, peptic ulcers, and cancer than matched general population controls. ccAP therefore appears to be a feature associated with the members of the population who are already less healthy.

## DISCUSSION

This retrospective data linkage cohort study aims to investigate the causes and predictors of mortality in the years following an episode of AP requiring critical care admission. In so doing, statistically significant differences in frequency of the causes of death have been demonstrated between the ccAP patient cohort and the general population. In addition, the results indicate that long-term

prognosis after a critical care admission for AP is influenced to a greater extent by age at the time of index AP admission and existing comorbidity than by specific features of the index AP episode. New-onset comorbidity, particularly diabetes, is more frequent following ccAP than in the general population. We acknowledge that ccAP patients may have additional diagnoses made because these individuals seek more frequent contact with healthcare and therefore have the opportunity to get diagnosed with comorbidities. Furthermore, comorbidities derived from the SMR01 data only reveal diagnoses made at the time of a hospital admission and are therefore an underestimate of the true population prevalence. From our analysis it is not possible to discern whether those individuals were destined to develop those comorbidities regardless of their episode of AP, especially given that the AP cohort is less healthy overall than the matched general population. A similar association between MODS and mortality has been demonstrated

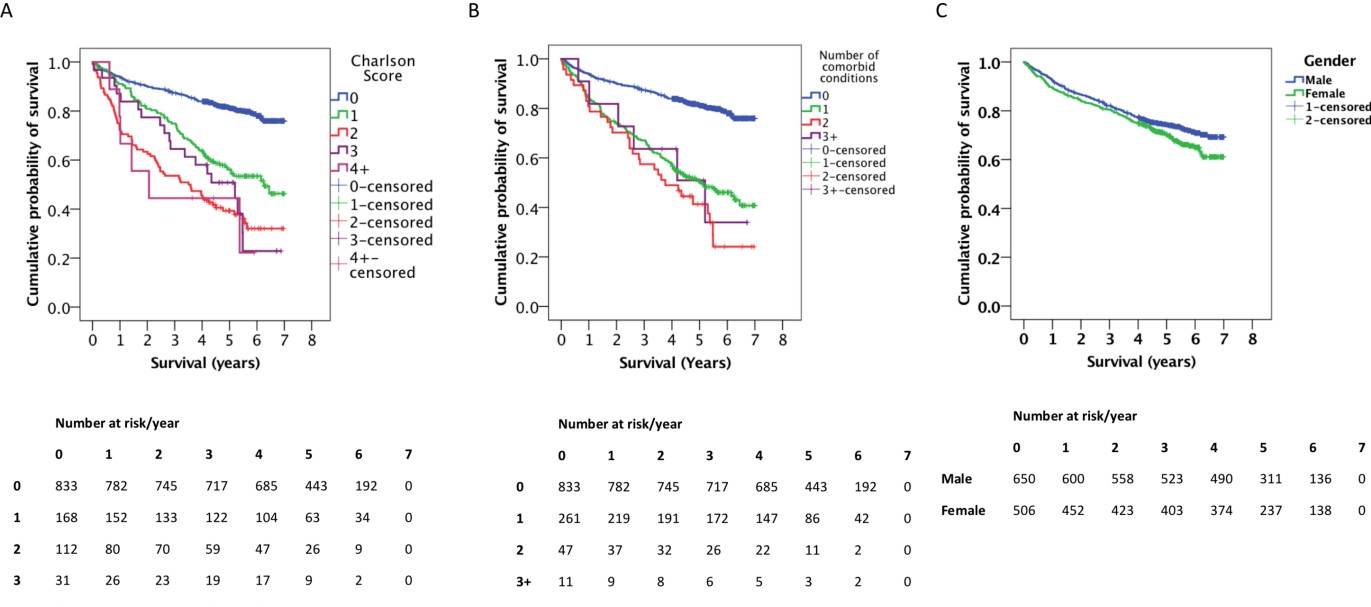

**Figure 3** Postdischarge survival of ccAP patients—patient characteristics. Kaplan-Meier survival plots for postdischarge ccAP patients, grouped by (A) Charlson score, (B) number of comorbid conditions contributing to each calculated Charlson score and (C) gender. The numbers of patients at risk at each time point are displayed. For each plot, in-hospital deaths have been excluded and time 0 corresponds to point of discharge. Vertical dashes represent right-censored patients. ccAP, acute pancreatitis requiring critical care admission.

in patients who have sustained trauma.[28] Our results support our previously observed concept that ccAP is associated with a persistent deleterious impact on survivors. Interindividual heterogeneity in the clinical course of the AP critical care episode was not associated with any organ-specific long-term outcomes in this analysis, but we acknowledge that our approach was limited in the ability to discriminate these with certainty.

Together, these findings lend weight to the hypothesis that severe AP episodes do not fully resolve, with particular emphasis on the impact of the associated systemic dysfunction. Our study has added data and analysis to underpin this concept by investigating the specific details of the deleterious legacy of ccAP. Given that the variation in causes of death is largely due to an

increased proportion of deaths from metabolic disease, it is reasonable to infer that AP mediates the long-term effects primarily through ongoing metabolic pathology. This result concurs with outcomes in a Danish cohort that demonstrated a marked increase in deaths from digestive system causes in AP survivors compared with the general population.[29] The exact mechanisms underpinning the metabolic disturbance remain to be elucidated and will almost certainly require a prospective experimental medicine study. Taken together, the reported high incidence of diabetes mellitus after AP, the correlation of AP severity with lasting pancreatic exocrine dysfunction (as shown by others) and the negative effect of exocrine pancreatic insufficiency suggest that impairment of the endocrine and exocrine pancreas is the main driver of the lasting overall dysfunction.[15 21 22] Additionally, it is reasonable to expect that aspects of the acute systemic dysfunction associated with MODS, for example, insulin resistance and mitochondrial dysfunction, fail to resolve entirely,[30] although we have not tested this experimentally in this study.

Identifying predictors of postdischarge mortality will facilitate appropriate targeting of preventative interventions. The identification of greater pre-existing comorbidity as a key negative predictive factor is consistent with previous research correlating more extensive comorbid disease with a worse prognosis after critical illness.[31] In the present study, our observation that postdischarge outcomes were better for ICU than HDU patients by univariate analysis could be explained by comorbidity—those requiring ICU admission theoretically experience the worst AP episodes, and therefore only relatively fitter

**Table 3** Final model of prognostic factors of postdischarge mortality

| Risk factor | HR | 95% CI | P value |
| --- | --- | --- | --- |
| Age* | 1.0 | 1.0 to 1.1 | <0.001 |
| Charlson score | 1.5 | 1.4 to 1.7 | <0.001 |
| Female gender | 1.2 | 1.0 to 1.5 | 0.058 |
| Gallstone aetiology | 0.7 | 0.6 to 0.9 | 0.003 |

The HR, 95% CI and p value of Wald's test are presented for each variable retained in the final multivariate Cox regression model. P<0.05 was considered significant.
Control variables not in the final model: renal replacement therapy, invasive ventilation, non-invasive ventilation, vasopressor use and Scottish Index of Multiple Deprivation.
*Age group as defined in table 2.

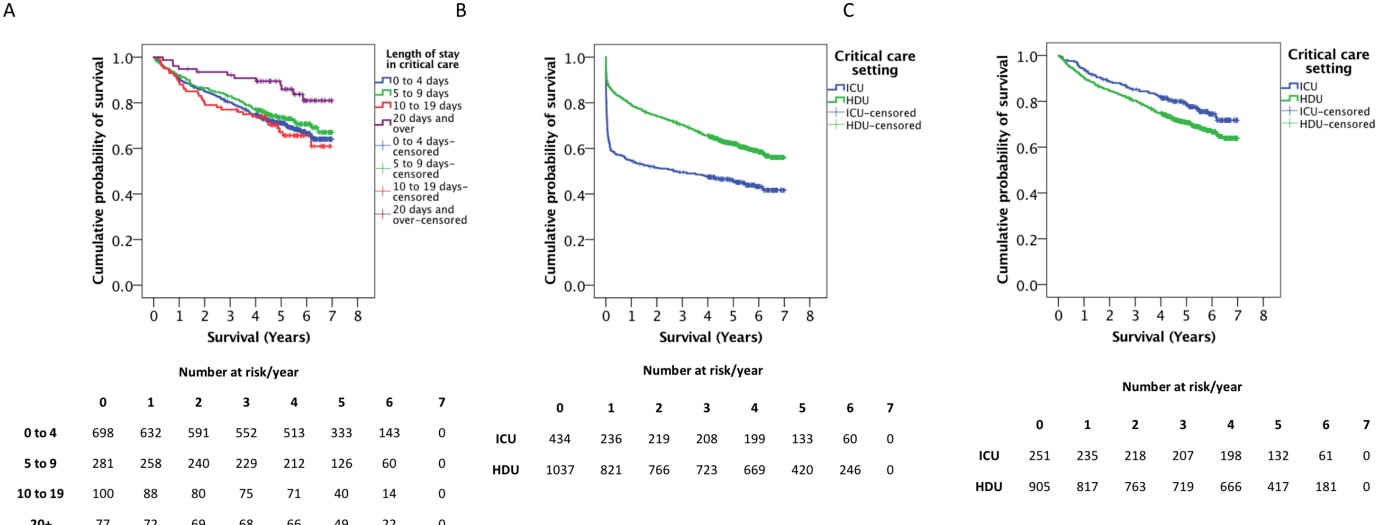

**Figure 4** Postdischarge survival of ccAP patients—nature of the critical care admission. Kaplan-Meier survival plots for postdischarge ccAP patients, grouped by (A) duration of critical care admission, (B) level of critical care (all ccAP patients) and (C) level of critical care (post-ccAP patients, excluding in-hospital deaths). The numbers of patients at risk at each time point are displayed for each plot. Vertical dashes represent right-censored patients. ccAP, acute pancreatitis requiring critical care admission; HDU, high-dependency unit; ICU, intensive care unit.

individuals may survive to discharge. In contrast, those with greater comorbidity, and hence a higher risk of later mortality, may survive an AP episode managed in

HDU. We acknowledge that the preceding statement is somewhat speculative, despite being highly plausible. A similar argument may explain the association of better

**Table 4** Baseline comorbid status and risk of developing new comorbidities after the index AP episode

| Comorbidity | Comorbidities at SICSAG admission | | | New comorbidities developed after discharge | | | | | |
| --- | --- | --- | --- | --- | --- | --- | --- | --- | --- |
| | % of AP cohort | % of controls | P value | % of AP cohort | % of controls | P value | OR | 95% CI lower | 95% CI upper |
| AMI | 2.7 | 1.2 | <0.001 | 3.7 | 1.8 | <0.001 | 2.09 | 1.4 | 3.12 |
| Cerebral vascular accident | 1.9 | 1.7 | 0.555 | 3.5 | 2.2 | 0.021 | 1.58 | 1.07 | 2.35 |
| Congestive heart failure | 2.3 | 0.4 | <0.001 | 2.5 | 0.8 | <0.001 | 3.11 | 1.83 | 5.28 |
| Connective tissue disorder | 0.8 | 0.3 | 0.024 | 0.1 | 0.1 | 0.641 | 0.6 | 0.07 | 5.16 |
| Diabetes and diabetes complications | 1.8 | 0.2 | <0.001 | 3.9 | 0.4 | <0.001 | 10.7 | 5.74 | 19.94 |
| Liver disease and severe liver disease | 0.5 | 0.1 | 0.004 | 0.6 | 0.1 | 0.003 | 5.3 | 1.55 | 18.14 |
| Peptic ulcer | 1.9 | 0.3 | <0.001 | 1.6 | 0.3 | <0.001 | 5.06 | 2.38 | 10.74 |
| Peripheral vascular disease | 1.5 | 0.6 | 0.007 | 1.3 | 0.5 | 0.002 | 2.86 | 1.41 | 5.81 |
| Pulmonary disease | 3.8 | 1.1 | <0.001 | 2.5 | 1.6 | 0.033 | 1.65 | 1.04 | 2.62 |
| Cancer and metastatic cancer | 7.2 | 2.8 | <0.001 | 8.4 | 5.4 | <0.001 | 1.62 | 1.25 | 2.11 |
| Renal disease | 1 | 0.2 | 0.001 | 1.1 | 0.1 | <0.001 | 9.15 | 2.95 | 28.43 |

The percentage of patients and controls who developed each specified comorbidity in the 5 years before and 5 years after the index AP episode is presented. The OR as well as the 95% CI for the development of each comorbidity after the AP episode are included. Total number of patients from the AP cohort: 1150; total number of controls: 3450. The p values were obtained by applying the two-sample z test. P<0.05 was considered significant.
AMI, acute myocardial infarction; AP, acute pancreatitis; SICSAG, Scottish Intensive Care Society Audit Group software; OR, odds ratio.

long-term outcomes with a critical care stay exceeding 20 days, in that individuals with less associated comorbidity at AP onset may be more resilient to a prolonged critical care admission. This finding is in contrast to data on long-term survival in all ICU patients, where prolonged admission was associated with a shorter long-term survival.[32 33] However, the positive association between duration of organ support in ICU and post-ccAP survival is likely subject to iatrogenic influences. For example, a willingness to persist with organ support in critical care by the physician-led multidisciplinary team in those without significant medical comorbidity prior to ICU admission may result in organ support being continued for longer, a form of survivor treatment selection bias.[34]

We observed that gallstone aetiology had a less negative effect on prognosis. While this might be explained by the additional burden of morbidity and mortality carried by alcohol misuse, the other key cause of AP,[35–37] our data imply that gallstone AP requiring critical care has less severe long-term consequences. This is in contrast to previous studies by others, where, in acute AP, a gallstone aetiology was associated with more severe MODS than alcohol-induced cases,[38] and separately no effect of gallstone aetiology on long-term prognosis after accounting for the detrimental impact of alcohol.[29] It is important to note that alcohol-related AP was not specifically known in our study population.

A strength of this study is that the applicability of these observations to all ccAP survivors has been enhanced by using primary data from a population basis rather than a single centre, in collaboration with Farr@Scotland. This UK-wide network was created to facilitate the storage, sharing and analysis of population and health-related data sets in an environment that protects patient confidentiality and data security.[39] The employment of this resource facilitated the achievement of larger sample population than would have been possible with a single-centre study and reduced the risk of the results being modified by, for example, regional variations in treatment or population demographics.[40 41] Matching each member of the ccAP cohort by year of birth, deprivation and sex to three individuals sampled from the remaining general population diminished any potential influence of national secular trends in the population incidence on the specific outcomes measured.

We acknowledge specific limitations of our study. First, the use of pre-existing national databases requires an acceptance of low amounts of missing data that might have been avoided had it been possible to prospectively capture all primary data. However, the expense and time needed to do that would make a study of this size extremely unwieldy, and we regard our approach to be preferable to that, at this stage. The incidence of missing data was very low, with the exception of APACHE II scores. In order to overcome data inaccuracies, only gallstone aetiology was specifically noted. Our experience of using these records to correctly attribute alcohol aetiology have not been sufficiently reliable as a foundation for a robust

analysis. Although not possible within the limitations of the current study, this will be an important consideration in advancing this research. Because there was uncertainty in our attribution of ccAP aetiology, with the exception of those diagnosed with gallstones, coupled with our use of relatively healthy controls from the general population, we were unable to analyse future causes of death and survival bias based on that factor. Insufficient detail in this data set precluded a robust analysis regarding the frequency of recurrent episodes of AP in the cohort, because it was not possible to distinguish repeat hospital admissions due to complications arising from the index episode from true de novo recurrent episodes. This would be addressed by a prospective study. Finally, the analysis of existing and new comorbidities was limited by relatively low proportions of patients affected by each comorbidity. The value of replicating these findings using larger patient cohorts would need to be weighed against the practical challenges but should be considered. Further clarification of this phenomenon, and the impact on other body systems, is in progress through a prospective experimental medicine cohort study.[42] The identification of specific goals for intervention in the follow-up period after AP will require that detailed assessment of alterations in patients' physiological status over time.

In conclusion, long-term outcomes after AP requiring critical care are influenced by pre-existing patient characteristics and specific factors associated with an episode of critical care admission. Persisting metabolic derangement after ccAP is associated with premature death. The persistent deleterious impact of severe AP on survival is multifactorial, and further mechanistic and epidemiological investigation is required.

**Acknowledgements** DJM acknowledges the support of the MRC through a Senior Clinical Fellowship. We are grateful to the members of APPLe for their personal insight and advise on our overall programme of work.

**Contributors** DJM, CS and CV conceived the study. Data retrieval, linkage and secure storage were done by DK and SN. Statistical design and analysis were done by SN and CG. CV and DJM drafted the initial version of the manuscript. All authors revised and approved the final version of the paper. DJM is the guarantor.

**Funding** This study did not receive specific funding. Data retrieval costs were covered by existing research funds. DJM currently holds a Medical Research Council Senior Clinical Fellowship (MR/P008887/1).

**Competing interests** None declared.

**Patient consent for publication** Not required.

**Ethics approval** Approval was obtained from the Privacy Advisory Committee of Information Services Division (ISD) Scotland. Research ethical committee review was not required for this study after consulting the guidance applicable to Scotland publicly available from the UK NHS Health Research Authority.

**Provenance and peer review** Not commissioned; externally peer reviewed.

**Data sharing statement** Original data are available on request through the NHS National Services Division Safe Haven, subject to an approval process. Further details may be found at http://www.isdscotland.org/Products-and-Services/eDRIS. Please contact the corresponding author in the first instance.

and indication of whether changes were made. See: https://creativecommons.org/licenses/by/4.0/.

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
