## [Reviewer comments · BMJ Open]

ARTICLE DETAILS

TITLE (PROVISIONAL)	Survival and new-onset morbidity after critical care admission for acute pancreatitis in Scotland: a national electronic healthcare record linkage cohort study.
AUTHORS	Ventre, Chiara; Nowell, Sian; Graham, Catriona; Kidd, Doug; Skouras, Christos; Mole, Damian

VERSION 1 – REVIEW

REVIEWER	Joan Casey University of California, Berkeley
REVIEW RETURNED	15-May-2018

GENERAL COMMENTS	This manuscript uses electronic health record data in Scotland to (1) identify comorbidities at time of an acute pancreatitis (AP) episode that required critical care admission and if they predict in-hospital death, (2) evaluate predictors of mortality after leaving the hospital, and (3) compare AP cases to population controls on comorbidities and causes of death. The authors provide many results with utility for different purposes. Their main finding was that sicker patients at ccAP survived for less time afterwards. They also observed increased new diagnoses for various comorbidities (including T2DM) in the ccAP population compared to population controls. The authors highlight the utility of EHR data to conduct these types of analysis. ##Major comments -The authors selected potential confounding variables to include in models using a data-driven approach. This resulted in just 3 variables making it into the final Cox model: age, Charlson score, and gallstone aetiology. This is surprising and may be an artifact of this particular data set. I would recommend selecting confounding variables also based on theory rather than arbitrary cutpoints like $p < 0.05$. At the very least, sex should be included in the final models. It also unclear in the text and in Table 3 which variables actually made it into adjusted models.-The authors conclude in the discussion: "Our results support the previously observed concept that ccAP leaves a persistent deleterious impact on survivors." I would soften this language. It's very possible (and perhaps probable) that ccAP and comorbidities like diabetes, renal disease, and peptic ulcer share common causes. If the authors could have compared patients with less severe AP to those with ccAP and found similar, but less severe comorbidity onset, I would be more convinced that ccAP causes these later outcomes.-ccAP patients may have additional diagnoses because they seek more healthcare and therefore have the opportunity to get diagnosed with comorbidities.
--

	-Additional limitations: inability to tell between common causes of ccAP and future causes of data, survival bias, extremely healthy population controls may not have been the best comparison population. -Table 4 suggests that just 0.2% of population controls had T2DM. A quick search showed that about 3.5% of the Scottish population has T2DM. Why the discrepancy? Should we trust these diagnostic data? -In Table 2, the hazard ratio indicates that women were less likely to die, but on Figure 3C women are depicted in green and appear to die sooner. Which is correct? Isn't Table 2 unadjusted? ##Minor comments Abstract -The setting is not the population of Scotland. Could you be more specific? -“ 340 of 1150 patients subsequently died.” Since everyone eventually dies, include the timeframe. -“ Median survival was 5.6 ± 0.1 years.” Consider removing this sentence from the abstract. It's misleading as follow-up lasted just 4.9 years (median after excluding patients who died in hospital). We don't really know how long folks would have survived as many were censored. -“Co-morbidity prior to ccAP negatively influenced survival” is vague. Possible to be more specific? i.e., More co-morbidities, including xyz, prior to ccAP negatively influenced survival in hospital and after discharge. Summary -“ 3. The use of pre-existing national databases resulted in low amounts of missing data. This might have been avoided had it been possible to prospectively capture all primary data.” These two sentences appear to contradict each other. You had little missing data and then you could have avoided it? Please re-write for clarity. Body -“ AP was defined as any admission with an ICD-10 diagnosis coded as K85 (acute pancreatitis).” Has this definition been validated? Did AP have to be the primary diagnosis code? If not, how many codes were used? If alternate definitions of AP were considered, please discuss. This would be a useful addition for future researchers looking to identify AP in EHR. - Were controls frequency matched on year of age or categories? Why were they not matched on year of diagnosis or encounter? It seems this could be a major confounder easily controlled with matching. -The second paragraph of the discussion is helpful. -The authors found that “long-term survival of those who survived the index episode was significantly better where the length of stay in critical care during the index episode exceeded 20 days, compared to admissions of 0-4 or 10-19 days.” This is intriguing and certainly points to a healthy patient survivor bias. Providing a couple of references to this type of bias in the discussion would be useful. -It is assumed the non-gallstone aetiologies are alcohol-related, but as noted alcohol-related AP is not specifically known in this study population. Please make this clear in the second paragraph on page 15.
--	---

	Tables  -For all tables provide more detailed titles and descriptions. The tables should stand alone from the manuscript. Also, define abbreviations in a footnote. -Add row percentages to table 1 -Add n in each category to Table 2 (this will help the reader interpret the very wide confidence intervals). -Table 3: Are these the only variables in the adjusted model? If not, add control variables in a footnote. What does “Age group” mean? This should be defined as the prior table breaks out the age groups used.
--	--

REVIEWER	Nisha Hazra King's College London
REVIEW RETURNED	22-May-2018

GENERAL COMMENTS	This is an important topic for AP survivors, using routinely collected data to evaluate and understand the predictors of mortality after an episode of AP requiring critical care admission. The authors state that preventing AP-MODS is an 'elusive goal' and therefore they want to characterise the lasting impact on survivors in terms of which organ deficits are associated with higher mortality or other poor outcomes to help maximise long-term well-being of AP survivors. General comments:  - Study objectives and design are a bit confusing. Authors state that this is a retrospective cohort study but then use a matched control group from the general population, making this seem a bit like a case-control study. - Sounds like the group of interest is those experiencing ccAP and focus is on associations between risk factors and mortality after AP (Aim in abstract: “To document organ-specific new morbidity and identify risk factors associated with premature mortality after an episode of ccAP”, and at end of introduction: “To investigate the causes and predictors of mortality in the years following an episode of AP requiring critical care admission”), rather than the difference between ccAP group and control group? If interested in outcome after an episode of AP, why is there a matched control group that did not have an episode of AP? Overall the objectives and design don't seem to fully align with each other. - Why did authors choose to use both the Charlson comorbidity index and number of comorbid conditions instead of using one or the other? - It is not detailed explicitly in the manuscript which specific variables are coming from which data source. This needs to be clearer for the study to be reproduceable. - What are the implications of the study conclusions? If many pre-existing characteristics (i.e. age) are most strongly associated with long-term outcomes after AP requiring critical care, and these are non-modifiable, what are the implications of this? This could be discussed more. Specific comments: Abstract:  - It is a bit odd to introduce the 'intervention' as 'record linkage of routinely-collected electronic health data with population matching'. This is more of a method than an intervention and not sure this study has an intervention.
--

	Introduction:  - I can assume that AP-MODS is associated with critical care admission (as mentioned in abstract) when reading the start of the introduction, but I wonder why AP-MODS is not mentioned in the abstract when there is so much focus on it in the introduction? Study Design, Data Security and Patient Confidentiality:  - Is the data anonymised? It says informed consent was not required/sought? Variables of interest:  - Some acronyms need to be defined and explained: APACHE II score, HDU, ICU. Even though most may know what ICU stands for this needs to be defined in the first instance in the manuscript - Not clear which variables were taken from which data sources, authors only state 'the following details were recorded' - this is too vague Statistical analysis:  - Proportional hazards assumption can also be tested using Schoenfeld residuals in addition to the visual check of the parallel curves Secondary Analysis of Associated Comorbidities:  - Authors state that comorbidities developed at admission and after discharge were obtained from acute hospital discharge records. But it is not clear whether these were taken from the Scottish Morbidity Records or the SICSA database. Where specific data came from (i.e. which database) could generally be described more transparently. Results:  - Under follow-up and survival, why could median survival time not be determined but the mean survival time could? - Authors mention difficulties in distinguishing 'repeat hospital admissions due to complications arising from index episode from true de novo recurrent episodes' (under development of new specific comorbidities, end of first paragraph) - firstly, this should probably be in the discussion section as a limitation, and secondly, if this means that it is difficult to determine the true index date then this might be a significant concern in the study design/methods which could affect the results of the study and also should be discussed - HDU vs ICU used in the univariate regression analysis but there is no description in the manuscript about the difference between or definition of HDU vs. ICU. Even if there is a clinical readership, cannot assume that all readers will know this. - I note that the results showed survival did not differ significantly with degree of social deprivation, but socioeconomic inequalities in mortality are well documented in the literature so this is a slightly surprising result. Can authors expand on why this might be? Discussion:  - The interpretation of better post-discharge outcomes in ICU compared to HDU explained by comorbidity seems like a bit of a stretch. Why would those with greater comorbidity and higher risk of later mortality survive an AP episode managed in HDU? Wouldn't those relatively fitter with less comorbidity be more likely to survive an episode in HDU too (as stated for ICU)?
--	--

	- last paragraph of pg 11: authors seem to acknowledge missing data as both a limitation (“acceptance of low amounts of missing data”) and a strength (“incidence of missing data was very low”) - was there any attempt to use multiple imputation? How is only noting gallstone aetiology a way to overcome data inaccuracies? - Authors acknowledge the interesting point that comorbidities are low in the cohort, but would be good to have some discussion around why they think this is?
--	---

VERSION 1 – AUTHOR RESPONSE

Reviewer: 1

1. The authors selected potential confounding variables to include in models using a data-driven approach. This resulted in just 3 variables making it into the final Cox model: age, Charlson score, and gallstone aetiology. This is surprising and may be an artifact of this particular data set. I would recommend selecting confounding variables also based on theory rather than arbitrary cutpoints like $p < 0.05$. At the very least, sex should be included in the final models. It also unclear in the text and in Table 3 which variables actually made it into adjusted models.

Sex has now been reintroduced to Table 3. A list of variables not entered is now provided in a footnote to Table 3.

2. The authors conclude in the discussion: “Our results support the previously observed concept that ccAP leaves a persistent deleterious impact on survivors.” I would soften this language. It’s very possible (and perhaps probable) that ccAP and comorbidities like diabetes, renal disease, and peptic ulcer share common causes. If the authors could have compared patients with less severe AP to those with ccAP and found similar, but less severe comorbidity onset, I would be more convinced that ccAP causes these later outcomes.

We agree this is reasonable. We have changed this sentence changed to “Our results support the previously observed concept that ccAP is associated with a persistent deleterious impact on survivors.”

3. ccAP patients may have additional diagnoses because they seek more healthcare and therefore have the opportunity to get diagnosed with comorbidities.

This is a good point, which we have now acknowledged in the discussion, page 9 para 1.

4. Additional limitation: Inability to tell between common causes of ccAP and future causes of data, survival bias, extremely healthy population controls may not have been the best comparison population.

We have acknowledged this in the discussion page 10 para 4.

5. Table 4 suggests that just 0.2% of population controls had T2DM. A quick search showed that about 3.5% of the Scottish population has T2DM. Why the discrepancy? Should we trust these diagnostic data?

This discrepancy most likely arises because the data in Table 2 are determined from SMR01 data i.e. people with a diagnosis of T2DM who have had a hospital admission during the the study period (and the 5 year look back period). This has been clarified in the text and the Tables. Specifically, we have added an additional limitation in the headlines, and also a sentence in the discussion, para 1: "6. Co-morbidities derived from SMR01 data only reveal diagnoses recorded at the time of a hospital admission and therefore are an underestimate of the true population prevalence."

6. In Table 2, the hazard ratio indicates that women were less likely to die, but on Figure 3C women are depicted in green and appear to die sooner. Which is correct? Isn't Table 2 unadjusted?

Thank you for spotting this error, due to reversing the reference category in Table 2. This has been corrected and the correct HR and 95%CI are now in the table.

7. The setting is not the population of Scotland. Could you be more specific?

This has been clarified to: "ccAP cohort: people admitted to critical care in Scotland; population cohort: community health index register of all patients in Scotland."

8. "340 of 1150 patients subsequently died." Since everyone eventually dies, include the timeframe.

This has been clarified. "340 of 1150 patients in the resulting post-discharge ccAP cohort died during the follow-up period"

9. "Median survival was 5.6 ± 0.1 years." Consider removing this sentence from the abstract. It's misleading as follow-up lasted just 4.9 years (median after excluding patients who died in hospital). We don't really know how long folks would have survived as many were censored.

This sentence now removed.

10. "Co-morbidity prior to ccAP negatively influenced survival" is vague. Possible to be more specific? i.e., More co-morbidities, including xyz, prior to ccAP negatively influenced survival in hospital and after discharge.

This sentence changed to: "Greater co-morbidity measured by the Charlson score, prior to ccAP, negatively influenced survival in hospital and after discharge."

11. The use of pre-existing national databases resulted in low amounts of missing data. This might have been avoided had it been possible to prospectively capture all primary data." These two sentences appear to contradict each other. You had little missing data and then you could have avoided it? Please re-write for clarity.

This sentence changed to: "The use of pre-existing national databases resulted in low, but not negligible, amounts of missing data. The amount of missing data might have been further reduced had it been possible to prospectively capture all primary data."

12. "AP was defined as any admission with an ICD-10 diagnosis coded as K85 (acute pancreatitis)." Has this definition been validated? Did AP have to be the primary diagnosis code? If not, how many codes were used? If alternate definitions of AP were considered, please discuss. This would be a useful addition for future researchers looking to identify AP in EHR.

This is now clarified: "AP was defined as any admission to critical care where the primary diagnosis coded by the intensive care senior clinician on duty was recorded as ICD-10 classification K85 (acute

pancreatitis).” The Wardwatcher database, at the time of the study, did not record second and subsequent diagnoses. No alternate definitions were used in our work. We used K85 as a handle to identify the episodes and retrieve the records.

13. Were controls frequency matched on year of age or categories? Why were they not matched on year of diagnosis or encounter? It seems this could be a major confounder easily controlled with matching.

Controls were matched on year of birth, deprivation quintile and gender of cases.

14. The second paragraph of the discussion is helpful. The authors found that “long-term survival of those who survived the index episode was significantly better where the length of stay in critical care during the index episode exceeded 20 days, compared to admissions of 0-4 or 10-19 days.” This is intriguing and certainly points to a healthy patient survivor bias. Providing a couple of references to this type of bias in the discussion would be useful.

We agree. This has been done.

15. It is assumed the non-gallstone aetiologies are alcohol-related, but as noted alcohol-related AP is not specifically known in this study population. Please make this clear in the second paragraph on page 15.

This has been clarified on page 10

16. Tables -For all tables provide more detailed titles and descriptions. The tables should stand alone from the manuscript. Also, define abbreviations in a footnote.

This has now been done.

17. Add row percentages to table 1

Row percentages are added.

18. Add n in each category to Table 2 (this will help the reader interpret the very wide confidence intervals).

This has been done.

19. Table 3: Are these the only variables in the adjusted model? If not, add control variables in a footnote. What does “Age group” mean? This should be defined as the prior table breaks out the age groups used.

A footnote with control variables has been added.

Reviewer: 2

1. Study objectives and design are a bit confusing. Authors state that this is a retrospective cohort study but then use a matched control group from the general population, making this seem a bit like a case-control study.

We understand why the reviewer raises the point, but we have followed the convention that if the study examines the disease and then evaluates the outcome, it is a cohort study, and if it starts with

the outcome, then looks for potentially causative factors it is a case-control study. Based on this we think that it is more correct to retain the existing nomenclature.

2. Sounds like the group of interest is those experiencing ccAP and focus is on associations between risk factors and mortality after AP (Aim in abstract: "To document organ-specific new morbidity and identify risk factors associated with premature mortality after an episode of ccAP", and at end of introduction: "To investigate the causes and predictors of mortality in the years following an episode of AP requiring critical care admission"), rather than the difference between ccAP group and control group? If interested in outcome after an episode of AP, why is there a matched control group that did not have an episode of AP? Overall the objectives and design don't seem to fully align with each other.

The aims of the study and the design are aligned, within the constraints of the routinely collected healthcare data available to us at the time of the analysis, and within pragmatic bounds. Including a non-critical care AP matched group would be good but would require an entire de novo record linkage study – which is a separate project in itself, and we do not propose to do that for this analysis.

3. Why did authors choose to use both the Charlson comorbidity index and number of comorbid conditions instead of using one or the other?

We did this because certain comorbidities within the Charlson index score more than 1 point (e.g. chronic liver disease), and we wished to provide the reader with as much information as possible.

4. It is not detailed explicitly in the manuscript which specific variables are coming from which data source. This needs to be clearer for the study to be reproducible.

This has been clarified in the manuscript.

5. What are the implications of the study conclusions? If many pre-existing characteristics (i.e. age) are most strongly associated with long-term outcomes after AP requiring critical care, and these are non-modifiable, what are the implications of this? This could be discussed more.

This is already present in the manuscript to a degree that we are comfortable with that would not constitute speculation.

6. It is a bit odd to introduce the 'intervention' as 'record linkage of routinely-collected electronic health data with population matching'. This is more of a method than an intervention and not sure this study has an intervention.

We have renamed this as "Methods".

7. I can assume that AP-MODS is associated with critical care admission (as mentioned in abstract) when reading the start of the introduction, but I wonder why AP-MODS is not mentioned in the abstract when there is so much focus on it in the introduction?

We have not done this for reasons of accuracy. Based on the detail of the routinely collected data, it would not be possible to prove that all the patients admitted to critical care met the definition criteria for AP-MODS. While nearly all will have done, we do not have the data readily available to prove that. We do know, however, that all those with ccAP were admitted to critical care, and therefore we can be certain about that. We do not wish to conflate definitions and therefore have retained the existing nomenclature.

8. Study Design, Data Security and Patient Confidentiality: Is the data anonymised? It says informed consent was not required/sought?

Correct, informed consent was not required. All data was linked and as far as possible personal identifiers were removed, and handled in the National Safe Haven according to the Charter for Safe Havens in Scotland. A reference to this has been added.

9. Variables of interest: Some acronyms need to be defined and explained: APACHE II score, HDU, ICU. Even though most may know what ICU stands for this needs to be defined in the first instance in the manuscript

This has now been done.

10. Statistical analysis: Proportional hazards assumption can also be tested using Schoenfeld residuals in addition to the visual check of the parallel curves

We agree that technically this could be done, but we did not think that it would simplify the manuscript, rather the converse, nor add further robustness to the conclusions we have drawn.

11. Secondary Analysis of Associated Comorbidities: Authors state that comorbidities developed at admission and after discharge were obtained from acute hospital discharge records. But it is not clear whether these were taken from the Scottish Morbidity Records or the SICSA database. Where specific data came from (i.e. which database) could generally be described more transparently.

This has been done.

12. Results: Under follow-up and survival, why could median survival time not be determined but the mean survival time could?

Because fewer than half of the cohort had died, as is stated in the manuscript.

13. Authors mention difficulties in distinguishing 'repeat hospital admissions due to complications arising from index episode from true de novo recurrent episodes' (under development of new specific comorbidities, end of first paragraph) - firstly, this should probably be in the discussion section as a limitation, and secondly, if this means that it is difficult to determine the true index date then this might be a significant concern in the study design/methods which could affect the results of the study and also should be discussed

This has been moved to the discussion as requested.

14. HDU vs ICU used in the univariate regression analysis but there is no description in the manuscript about the difference between or definition of HDU vs. ICU. Even if there is a clinical readership, cannot assume that all readers will know this.

These are standard definitions for which a reference has been provided.

15. I note that the results showed survival did not differ significantly with degree of social deprivation, but socioeconomic inequalities in mortality are well documented in the literature so this is a slightly surprising result. Can authors expand on why this might be?

This is interesting but would be speculation and not something we could comment on based on our data.

16. The interpretation of better post-discharge outcomes in ICU compared to HDU explained by comorbidity seems like a bit of a stretch. Why would those with greater comorbidity and higher risk of later mortality survive an AP episode managed in HDU? Wouldn't those relatively fitter with less comorbidity be more likely to survive an episode in HDU too (as stated for ICU)?

The latter group are less likely to be admitted to critical care at all. However, we don't have that data, so we can't speculate in the manuscript.

17. last paragraph of pg 11: authors seem to acknowledge missing data as both a limitation ("acceptance of low amounts of missing data") and a strength ("incidence of missing data was very low") - was there any attempt to use multiple imputation? How is only noting gallstone aetiology a way to overcome data inaccuracies?

This has been clarified. See response to reviewer 1, point 11.

18. Authors acknowledge the interesting point that comorbidities are low in the cohort, but would be good to have some discussion around why they think this is?

See response to reviewer 1 point 5 above.

VERSION 2 – REVIEW

REVIEWER	Joan Casey University of California, Berkeley, USA
REVIEW RETURNED	10-Jul-2018

GENERAL COMMENTS	The authors were very responsive to reviewer comments. -I have a lingering concern about the lack of adjustment for calendar year. While the authors match on age, they do not control for calendar trends in their data. One patient could be 75 in 2014 and another 85, but they will experience similar diagnosis trends or other aspects of Scotland at that time that might confound the primary associations of interest. Possible to check to see if adjusting models for calendar year makes any difference in associations? -Please add parenthetical age groups to the sentence: "Controls were frequency matched on deprivation quintile, age and sex."
---

VERSION 2 – AUTHOR RESPONSE

Reviewer comments:

1. I have a lingering concern about the lack of adjustment for calendar year. While the authors match on age, they do not control for calendar trends in their data. One patient could be 75 in 2014 and another 85, but they will experience similar diagnosis trends or other aspects of Scotland at that time that might confound the primary associations of interest. Possible to check to see if adjusting models for calendar year makes any difference in associations?

While we understand why the reviewer raises this point, because each member of the ccAP cohort was matched by year of birth to 3 individuals sampled from the general population, secular trends in diagnosis will apply equally to all individuals and we therefore have not re-run the analysis. We have included a statement to this effect in the discussion (p14 last para).

2. Please add parenthetical age groups to the sentence: "Controls were frequency matched on deprivation quintile, age and sex."

Controls were frequency matched by year of birth, rather than age group. This has been clarified by adding this detail in the methods p7 para 1.

Once again, we thank all the reviewers for their constructive input, and we hope that these further revisions to the manuscript make it suitable for publication.